# The Nature of Micro-Variability in Blazars

James R. Webb [1,*], Viviana Arroyave [1], Douglas Laurence [2], Stephen Revesz [1], Gopal Bhatta [3], Hal Hollingsworth [4], Sarah Dhalla [5], Emily Howard [2] and Michael Cioffi [1]

[1] Physics CASE Faculty, Florida International University, Miami, FL 33199, USA; varro008@fiu.edu (V.A.); sreve003@fiu.edu (S.R.); mciof005@fiu.edu (M.C.)
[2] STEM, Broward College, Fort Lauderdale, FL 33314, USA; dhlaurence@gmail.com
[3] Institute of Nuclear Physics Polish Academy of Sciences, 31342 Kraków, Poland; gopal.bhatta@ifj.edu.pl
[4] School of Science, Miami Community College, Miami, FL 33132, USA; hollingsworth@Miamicollege.edu
[5] STEM, Seminole State College, Orlando, FL 32773, USA; sarahdhalla@gmail.com
* Correspondence: webbj@fiu.edu

**Abstract:** We present the results of a long-term study designed to investigate the nature of micro-variability in blazars carried out primarily at the Southeastern Association for Research in Astronomy (SARA) observatories. We analyzed micro-variability data of fifteen OVV quasars and BL Lac sources collected from 1995 to 2021. The data set consists of single-band light curves interspersed with multi-color and micro-variability observations. This paper reports over 900 nights of CCD observations. We also incorporated observations from other observers as well as observations gleaned from the literature into our analysis. We employed differential photometry to measure magnitudes and then construct the long-term and micro-variability light curves. Our results indicate that there is no correlation between the presence of micro-variations and the brightness of the source. We present a viable theory to explain the intermittent micro-variability as pulses of radiation emitted by individual turbulent cells in the relativistic jet, which are stimulated by a passing shock wave. We present model fits and test results for various data sets, including WEBT light curves, Kepler light curves and a TESS light curve. Although the consensus in the community is that blazar jets must be turbulent, the identification of micro-variations as manifestations of actual turbulent cells is important for modeling these turbulent jets. We can obtain estimates of cell sizes (assuming a shock speed), and the distribution of cell sizes derived from observations is consistent with numerical simulation predictions.

**Keywords:** blazars; quasars; micro-variability; optical variability

**PACS:** J0101

## 1. Introduction

Blazars exhibit variability over a wide range of timescales. Long-term variability over timescales of years is seen in most blazars [1], and short-term outbursts with amplitudes of several magnitudes over timescales of weeks to months are frequently observed in most sources. Micro-variability, which is commonly defined as extremely rapid (t < 1 h) with low amplitude (amp < 0.1 mag) variations, is seen intermittently in blazars and BL Lac objects. Blazar micro-variability was shown to be intrinsic [2] and has been studied in detail for over fifteen years now ([3–7]). Fourier transform analysis, unequal interval periodic analysis, and wavelets failed to find significant periods, patterns, or noise components (1/f or red noise) [8,9].

The presence of turbulence in blazar jets was proposed by [10] in order to explain the lack of polarization seen in blazar radio jets. Jones [10] proposed that the synchrotron emission emitted by a turbulent magnetic field in the jet would be depolarized.

More recently, Ref. [11] developed a multi-zone turbulent jet model involving shocks to explain gamma-ray emission from blazar jets. Although simulations of turbulence in

relativistic jets are still in their infancy, Ref. [12] showed that many of the properties found in non-relativistic turbulent jets are similar to the properties in relativistic flows to first order. If we assume that micro-variations are manifestations of the turbulent nature of the jet flow, we can begin to use micro-variability to study the physics of the flow. If the object shows optical micro-variability, then according to this model, the shock is encountering large well-developed turbulent regions of the jet. If the jet flow is laminar or the turbulent size scales are too small at the location of the shock, then we will not see micro-variations in the light curve. Simulations show that relativistic turbulence under the conditions expected in blazar jets is by nature an intermittent phenomenon [13]. The duty cycle is a measure of how close to critical the flow is. The Duty Cycle (DC) is defined as the time the source shows micro-variability divided by the total observation time. If the duty cycle is large, 95%, then the jet is mostly well-developed turbulence, and the Reynolds number is much larger than critical (Re >> 2000). If the duty cycle is moderate, then the turbulence is not well developed with no large-scale eddies. The significance of the duty cycle to our model is the larger the duty cycle, the larger the Reynolds number means that well-developed turbulence is more likely. Observationally, the larger the duty cycle, the more likely we will observe variations we can model.

Although there is no way to predict when you would see a particular cell size, there is a range of size/time scales for the vortices. The smallest vortex length is normally associated with the Kolmogorov scale (where most of the dissipation takes place in a non-relativistic plasma), and the largest length scales are associated with either the size of the plasma jet or the correlation length within the plasma. Relativistic simulations show that turbulent relativistic extragalactic jets show a similar relationship between vortex length scale and energy as non-relativistic flows [12]. We collect micro-variability curves and deconvolve the curves into individual pulses. The analysis of these pulses can lead to an estimation of the sizes and numbers of the large turbulent vortices in the jet. There will undoubtedly be pulses from numerous smaller vortices that our photometry and time resolution cannot identify. Marscher estimates that there are on the order of 300 turbulent cells from the frequency and time dependence of linear polarization in blazar jets [14].

We present the observational program in Section 2, including instrumentation, target objects, and observing statistics. The resulting long-term light curves are presented in Appendix A. Section 3 describes the individual objects involved in this study in detail based on our observations. In Section 4, we consider correlations between micro-variability and flux level and calculate the duty cycles for each of the sources in our program. Section 4 also presents the model we have developed from this long-term study of micro-variability. This model assumes that the micro-variations are the result of a shock wave propagating down a turbulent jet and energizing individual turbulent cells that cool by emitting synchrotron radiation in the form of pulses [15]. The observed micro-variability curves would then be convolutions of these individual pulses, not just random noise. We fit the calculated pulse shapes to several of our micro-variability observations and then estimate the turbulent cell parameters by deconvolving these pulses. We also test the model for predicted spectral changes.

## 2. Observational Program

We routinely monitored a list of fifteen AGN, mostly blazars, three or four nights per month using the SARA 0.9-meter telescope since 1995 [16]. This program has been responsible for the detection of a number of outbursts, including major outbursts in PKS 1156+195 [17] and in BL Lac [18]. The primary goal of this program was the compilation of a large number of short-term and micro-variability observations over twenty years. We measured the preliminary magnitudes of each source in real time to determine the brightness of the source. We then followed up with a series of *VRI* images, and if the conditions were appropriate, we made micro-variability observations of the source. A typical micro-variability observation would consist of between 100 and 400 consecutive CCD images in a single filter in rapid succession. Exposure times range between 30 s for

bright sources and 4 min for fainter objects such that the exposure time was consistent with signal to noise ratio (S/N) that would yield 1% photometry. The comparison sequences from the Heidelberg blazar website [19] were used in each case to calibrate magnitudes, and care was taken to monitor tracking errors, FWHM variations, and the presence of clouds that might affect the photometry. Our program also contributed observations to numerous satellite backup and multi-wavelength campaigns [15,20,21].

The definition of micro-variability commonly used in the literature is simply "rapid variations on the order of tenths of a magnitude over time scales of minutes or hours". This definition does not distinguish between a constant linear trend throughout the night and low amplitude oscillations around either a constant or linearly increasing or decreasing background. A more precise definition of micro-variations would be: *brightness changes of 0.001–0.01 magnitudes over timescales of hours or minutes either above or below a linear background sampled over the entire night.* Within our new definition, linear increases or decreases in flux would not constitute micro-variability. The Howell statistical method [22] (hereafter "Howell Test") is used to determine the presence of real variations versus statistical noise by comparing the variances of the object to the variances of a comparison star and of a check star, taking into account the instrumental noise properties of the detector and the brightness differences between the source and the comparison stars. Then, based on the CCD characteristics (gain, aperture size. Background noise), it computes the probability that the variation seen in the variable is real and not random noise. We tested the resulting micro-variability light curves for statistically significant variability using the Howell Test. If significant variations were present in the light curve, we would conduct a linear least-squares fit of the data and subtract the linear trend. We then re-ran the Howell test on the linearized data. If the test again yielded a positive result, we would confirm the presence of micro-variations in the data. If the test was positive for the original light curve and negative for the detrended data, then we would conclude that micro-variations were not present in the data.

Table 1 shows the sources monitored as part of our SARA program along with the classification, redshift, and the number of nights we observed the object. Column 5 lists the number of long (3 to 8 h) micro-variability light curves obtained for that object, and column 6 lists the correlation bi-serial coefficient between the presence of micro-variability and the source brightness. The bi-serial correlation coefficient is used when one variable is dichotomous, whether micro-variability was detected or not. $R_{pb}$ is 0 if the data are uncorrelated and 1 for complete correlation. Listed in parenthesis is the resulting probability of the 2-tailed significance test.

In the data set as a whole, there was an obvious trend in observing the brighter blazars for micro-variability more frequently. Weather considerations such as high cirrus clouds or poor visibility, the brightness and accessibility of the source, as well as the sensitivity of the telescope/CCD combination determined if micro-variability observations could be made. We tried to make micro-variability observations of all of the sources over their complete brightness range based on previous long-term monitoring observations [1]. We concluded, based on the statistics and our new definition, that micro-variations in each object are intermittent with varying duty cycles. Previous results indicated that there was no correlation between the brightness level of the source and the presence of micro-variability for four objects included in this study [4]. Our goal was to test this result by making micro-variability observations over the entire range of observed optical variability for each object in Table 1. We reached a similar conclusion with all of the sources; there is no correlation between brightness level and the presence/absence of micro-variability.

**Table 1.** The SARA Micro-Variability Results.

| Object | Type | Redshift | Nights Observed | Micro Runs | Correlation Coefficient |
|--------|------|----------|-----------------|------------|-------------------------|
| AO 0235+164 | BL Lac | 0.940 | 88 | 15 | −0.48 (0.059) |
| PKS 0420-01 | OVV Quasar | 0.916 | 36 | 10 | −0.49 (0.321) |
| S5 0716+71 | BL Lac | 0.300 | 252 | 87 | −0.26 (0.016) |
| 0735+17 | BL Lac | 0.424 | 21 | 2 | — |
| 0736+017 | OVV | 0.189 | 7 | 3 | — |
| OJ 287 | BL Lac | 0.306 | 66 | 22 | +0.02 (0.926) |
| PKS 1156+295 | OVV | 0.725 | 57 | 10 | −0.4 (0.246) |
| 3C 273 | QSO | 0.158 | 14 | 4 | — |
| 3C 279 | OVV | 0.536 | 63 | 19 | −0.48 (0.039) |
| 1510-01 | OVV | 0.360 | 18 | 3 | — |
| 3C345 | OVV | 0.593 | 56 | 11 | +0.09 (0.785) |
| ON 231 | BL Lac | 0.102 | 13 | 6 | −0.2 (0.709) |
| 3C 454.3 | OVV | 0.859 | 10 | 6 | +0.56 (0.245) |
| BL Lac | BL Lac | 0.068 | 195 | 53 | +0.04 (0.773) |
| 3C 446 | OVV | 1.404 | 31 | 2 | — |

The long-term R-band light curves of each object in this study are presented in Figures A1–A15. The black points connected by a dotted line indicate monitoring observations on that night. The dotted line was included only to show the causal connection between data points and not to interpolate between data points. We routinely made at least five exposures per filter, sometimes more, and averaged them to determine the brightness of the object during normal monitoring. We made full-color sequences of VRI for each source on many nights, but only the R magnitudes are plotted in the figures. The micro-variability observations are denoted by green or red symbols on the plots.

A micro-variability observation consists of at least thirty and sometimes more than 400 individual CCD images in a single night. Long micro-variability light curves extend up to 10 or 12 h while shorter ones may only be 3 h long. Each individual image was analyzed, and a single night light curve with a time resolution of a minute to a few minutes was created. The green up-pointing triangle symbols indicate that a micro-variability observation was made that night and micro-variability was detected, while a red down-pointing symbol indicates that the micro-variability observation was made, but no statistically significant micro-variations were detected that night.

We briefly discuss each source in our list and comment on the activity level that we observed.

### 3. Sources

*3.1. AO 0235+164*

This BL Lac source can become quite faint, dropping below R = 20 magnitudes, so high signal-to-noise micro-variability observations are difficult to obtain using 1-m class telescopes when it is faint (see Figure A1). It did, however, go through extremely active periods and was seen as bright as R = 15.0. Our data cover the period between 1999 and 2021 and consists of 88 nights of observations. We detected a brightness range from R = 15.4 ± 0.03 down to R = 19.9 ± 0.6. The Johnson R filter was used most nights for normal monitoring and initially for micro-variability observations. This object showed micro-variability on four of the 15 micro-variability observations with a duty cycle of 22.42% (see Section 4 for the definition of duty cycle). The presence of micro-variability is not correlated with brightness, as seen from the correlation coefficient of −0.48. Multi-color

sequences were made on 21 nights in three filters (VRI) which allowed us to calculate and compare the average color index R-I = 1.104 with a standard deviation of 0.345. There was no apparent correlation between the spectral index and R magnitude in the data. The average magnitude of AO 0235 + 164 during this time was R = 17.8 over a fourteen-year period. Our observations are consistent with the long-term light curve accumulated from the Tuorla Blazar monitoring program [23], showing major outbursts at the same time. The Tuorla group observed the two major outbursts in 2006 and 2008, reaching much brighter magnitudes (R = 14.2); however, we did not have a dense sampling of observations during those times and could not confirm.

Many groups have attempted to find periods in the light curve of this source [24,25], and also non-standard causes for variability such as gravitational lensing have been considered [26].

Periodicity searches such as [25] found evidence for a 5.7-year period for major outbursts in the long-term light curve, but a predicted outburst based on this period failed to materialize at the proper time [27]. None of the different data sets seemed to yield the same periods, so it is not clear that any strict periodicities actually exist in the long-term light curve of this object. Our data yield a maximum flux change of R = 0.193 magnitudes/day or 0.008 magnitudes/hour, while the most rapid reported flux increase was 0.056 mag/h [28].

### 3.2. PKS 0420-01

We observed PKS 0420-01 on thirty-six nights. The light curve is presented in Figure A2. Multi-color observations were made on several nights, and it was monitored for micro-variability on ten different nights. The light curve shows activity over a 6.4 magnitude range from R = 12.5 to R = 18.9. One major outburst was detected in December of 2002, and other minor outbursts occurred in November of 2003 and in January 2010. Micro-variability observations were made over the range of 14.8 to 18.03. Micro-variability was detected only once with the linear trend subtracted, giving it a very low duty cycle of 12.83%.

Many of the observations reported here were used in previous studies in an effort to find repeatable periodicities in the micro-variability. We analyzed the micro-variations in a number of sources, including PKS 0420-01, PKS 0736+01, PKS 1510-08, 3C 273, ON 231, and BL Lac [8]. They reported periods of 2.4 h and 1.413 h in micro-variability curves for 0420-01 by using unequal interval Fourier Transform (FT) analysis. Variability timescales of 0.12 magnitudes in 40 min were reported in 1997 [29], and variations of 0.083 magnitudes in 16.5 min were reported [30]. Both of these estimates yield a maximum rate of flux rise of about 0.005 magnitudes per minute from different data sets. Our average calculated maximum rise time over a night is 0.011 mags/hour, roughly twice those reported. Table 1 shows that the correlation between brightness and the occurrence of micro-variability is very small, r = +0.49. Figure A2 shows that the red-no micro-variability points span nearly the entire range of brightness, while the single micro-variability point lies very near the average brightness level for the source.

### 3.3. S5 0716+71

Blazar 0716+71 is one of our most frequently observed BL Lac objects and has been the target of numerous multi-frequency campaigns [3,5,25,31–35]. We observed S5 0716+71 252 times, and it varied from R = 12.28 to 14.09. Eighty-seven of these observations were micro-variability observations. These micro-variability observations nearly encompassed its entire range of variability. Table 2 shows the variability results for 0716+71. Twenty-nine observations clearly showed micro-variability while fifty-eight did not. Figure A3 shows that micro-variability was seen at all brightness levels, as was the lack of micro-variability. The duty cycle with linear trend removed was about 86%. This object showed the highest duty cycle of our sample. Table 1 shows that there is no correlation between the presence of micro-variability and brightness, which is strongly supported by the computed point

bi-serial correlation coefficient of −0.26 and the probability of 0.016 that the variables are correlated.

**Table 2.** Duty Cycle Results.

| Object | With Linear | DC (ROM) | DC | Without Linear | DC (Rom) | DC |
|---|---|---|---|---|---|---|
| AO 0235+164 | 9 Yes, 6 No | 55.23 | 35.57 | 4 Yes, 11 No | 47.14 | 22.42 |
| PKS 0420-01 | 2 Yes,4 No | 32.22 | 35.14 | 1 Yes, 5 No | 20.46 | 12.83 |
| S5 0716+71 | 54 Yes, 33 No | 44.89 | 77.40 | 29 Yes, 58 No | 17.32 | 51.81 |
| 0735+17 | 0 Yes, 2 No | 0.00 | 0.00 | 0 Yes, 2 No | 00.00 | 00.00 |
| 0736_017 | 3Yes, 0 No | 100.00 | 100.00 | 1 Yes, 2 No | 22.82 | 45.15 |
| OJ 287 | 14 Yes, 7 No | 48.79 | 79.00 | 3 Yes, 18 No | 7.65 | 19.81 |
| PKS 1156+295 | 5 Yes, 4 No | 58.66 | 60.12 | 5 Yes 5 No | 47.46 | 52.12 |
| 3C273 | 1 Yes, 3 No | 28.39 | 14.53 | 0 Yes, 4 No | 0.0 | 0.0 |
| 3C 279 | 8 Yes, 11 No | 39.18 | 36.81 | 7 Yes, 12 No | 34.56 | 33.50 |
| 1510-01 | 0 Yes, 3 No | 0.0 | 0.0 | 0 Yes, 3 No | 0.0 | 0.0 |
| 3C 345 | 5 Yes, 6 No | 44.85 | 49.48 | 2 Yes, 9 No | 25.42 | 14.84 |
| ON 231 | 3 Yes, 3 No | 24.61 | 58.28 | 1 Yes, 5 No | 8.25 | 17.43 |
| 3C454.3 | 4 Yes, 2 No | 67.74 | 69.06 | 3 Yes, 3 No | 44.02 | 58.82 |
| BL Lac | 20 Yes, 11 No | 56.67 | 72.59 | 13 Yes, 18 No | 39.61 | 54.73 |
| 3C 446 | 0 Yes, 2 No | 0.0 | 0.0 | 0 Yes, 2 No | 0.0 | 0.0 |

Wavelet analysis was performed on eight micro-variability curves taken between January 2004 and February 2005 and found no repeatable timescales in any of the micro-variability data [9]. Time series analysis of similar light curves was performed by a number of other groups and these analyses failed to yield any timescales that were repeated over different epochs or seen in different objects. These results cast doubt that there is a repeatable characteristic timescale embedded in the micro-variability data; the only periods above the noise were due to the finite length and imperfect sampling of the data. Although there are no repeatable periodicities in the sense that individual, micro-variability observations show essentially the same range of timescales, and there appears to be some structure to the fluctuations, they are not entirely white noise. Although the light curve amplitude and timescales vary without periodicities, the variations seem to have a common shape. A more recent WEBT observation [21] again displayed intense micro-variability as well as polarimetry variability [36]. These observations yielded maximum rise and decline rates of about 0.09 magnitudes/hour. This observation yielded more of the same: not strictly noise, but no periodicities.

*3.4. PKS 0735+17*

This object was part of a long-term variability study of BL Lacs and showed long-term trends of two magnitudes with shorter outbursts of 1.5 magnitudes [1]. A multi-frequency study of this source was published in [37]. This source was included in an intensive long and short-term variability study and exhibited about the same range of variability as reported in [1]. In our current study, we observed it on fifteen occasions and only looked for micro-variability on two occasions. The light curve is presented in Figure A4. Neither of the micro-variability runs yielded significant micro-variations. The light curve showed significant variability over a 1.7 magnitude range (14.27–15.96); however, both micro-variability observations were made at intermediate magnitudes (R = 15.26 and R = 15.57). The brightest magnitude was R = 14.277 (0.002) on 12/28/2008.

### 3.5. PKS 0736+071

This object is an OVV blazar at a redshift of 0.189 and was rarely observed in our program. The seven nights of observations included three attempts to observe micro-variability. Application of the Howell statistical method indicated initially that all three micro-variability observations showed micro-variability; however, only one light curve showed micro-variability after the linear trend was removed. The long-term light curve is presented in Figure A5.

### 3.6. OJ 287

This BL Lac object was regularly observed throughout the 1980s [1,38] and showed extreme short-term variability, as can be seen in Figure A6. We observed it 66 times in our program; on 22 of those nights, we looked for micro-variability. Initially, 14 observations tested "Yes" for micro-variability, 7 tested "No". Removal of the linear trend resulted in 3 "Yes" and 18 "No". This resulted in a duty cycle of about 19.81%. Once again, the presence of micro-variability was spread over the entire range of variation and appeared independent of brightness, as confirmed by the statistical tests in Table 1 $r_{pb}$ = 0.002.

### 3.7. PKS 1156+295

One of the most interesting and variable objects seen, PKS 1156+295, debuted on the quasar scene with a bang, undergoing a three magnitude (B~16 to B~13) optical outburst back in 1980 [37]. Figure A7 shows the light curve for this source. It recently underwent an outburst where it attained the brightest magnitude ever recorded for it. We observed it on 57 nights, and the variability range was from R = 14 to R = 18, a four magnitude range. We looked for micro-variability on ten nights, and five nights showed statistically significant micro-variability, while five nights did not. The resulting duty cycle is 52.12%.

### 3.8. 3C 273

Since 3C 273 rarely shows explosive brightness changes, we did not observe it frequently. It showed almost consistent fading in our observations during this period. Of the four micro-variability runs, none exhibited statistically significant micro-variability. Due to the lack of observations, the correlation coefficient in Table 1 was not calculated. The duty cycle listed in Table 2 is also not very reliable since it is based on only four micro-variability observations. The light curve is presented in Figure A8.

### 3.9. 3C 279

This is one of the most explosive blazars we have observed, and observations of it range over four magnitudes [20,39]. We observed it on 63 nights, and micro-variability observations were performed on 19 of those nights. It showed micro-variability 7 times while not exhibiting rapid variations 12 times. The data yielded a correlation coefficient of −0.58, with only a 4% chance that micro-variability is correlated with brightness level. The duty cycle was 33.5%. The light curve is shown in Figure A9.

### 3.10. PKS 1510-089

Although we rarely observed this object (18 observations), we looked at it three times for micro-variability. It never showed micro-variability, so no duty cycle is evident. See Figure A10.

### 3.11. 3C 345

One of the most well studied OVVs, 3C 345, regularly shows dramatic multi-frequency outbursts [1]. Fifty-six observations were made of this object; 19 of them were full-night micro-variability observations. This object showed one of the strongest correlations with flux level at $r_{pb}$ = +0.48. For the light curve, Figure A11 shows that micro-variability was seen at all levels of brightness except for the extremely bright point at JD 2452100, where no micro-variability was observed. The duty cycle is 33.5%.

### 3.12. ON 231

The observation of ON 231 first reported by [40] showed dramatic micro-variations that exhibited structure well above the noise. The resulting light curve was analyzed extensively using unequal interval FT and periodograms by [8] with inconclusive results. This particular light curve exhibited "pulse-like" characteristics that eventually led to the model presented in [15] and discussed in detail below. This object was observed only 13 times with 6 micro-variability runs, with only one micro-variability detection and a duty cycle of 17.43%. The light curve is shown in Figure A12.

### 3.13. 3C 454.3

This blazar was first noted to vary by Alan Sandage in 1966 [41]. Figure A13 shows our observations of 3C 454.3, which cover a two-magnitude range. Two observations made in 2001 showed it at a low of R = 15.5. It showed micro-variability at that low brightness level even with the linear trend subtracted. It was once again observed in 2005 and was shown to be undergoing a two magnitude outburst up to a maximum of R = 13.353 (0.005). It varied over a one magnitude range in June of 2005, showing a variation of 0.2 magnitudes in 24 h. During that month-long outburst, it showed micro-variability on 16 and 29 June and 17 October. A WEBT multi-frequency paper reported a relatively faint state R ~ 16.5 in 2007 [42], and it brightened up to R = 12.97 in 2014 [43]. It exhibited the highest duty cycle of all of our objects, DC = 58%. The correlation coefficient was +0.56. Figure A13.

### 3.14. BL Lac

BL Lac is well placed in the sky and very bright for a blazar, making it a very popular target for micro-variability observations for northern hemisphere telescopes [44]. Between 1971 and 1986, it varied almost continuously between B~11 and B~15, exhibiting frequent two magnitude flares and anti-flares [1]. Fan et al. [45] compiled the light curve from 1970 to 1996, showing its B magnitude ranging from 17.99 to 12.68 during this period of time. We observed it over 195 nights. We made 53 micro-variability observations between 1999 and 2021 with SARA North. During this time, BL Lac varied almost continuously, showing numerous rapid outbursts over a 2.5 magnitude range. As clearly seen in Figure A14 we have sampled BL Lac over its entire 2.5 magnitude range of variability, and it sometimes showed micro-variability when bright, sometimes when faint, thus with no apparent correlation with brightness $r_{pb}$ = +0.04. Because of its bright apparent magnitude, the statistics for determining the presence and absence of micro-variability are very significant. Note that it showed no micro-variability at its brightest (R ~ 12.5) and at its faintest (R ~ 14.7). The most rapid rise occurred in 1997 when it rose from R ~ 14.14 to R ~ 13.378 in just one day, giving a rise of 0.8 magnitudes per day. In an earlier study, no evidence was found for a correlation between the presence of micro-variability and brightness [4]. Howard et al. [4] further tested the correlation of micro-variability with the slope of the light curve and found a small correlation with the slope in the sense that positive micro-variability was more highly correlated with a steeper slope. The light curve for BL Lac is shown in Figure A14.

### 3.15. 3C446

Quasar 3C 446 is a powerful radio source and has shown rapid high amplitude outbursts since it was first observed. It tends to be relatively faint and thus was not frequently tested for micro-variability; there were only two micro-variability observations because of our moderate aperture telescopes. The light curve for 3C 446 is shown in Figure A15.

## 4. Observational Results

Micro-variations are intermittent; the duty cycle (*DC*) is defined as the percentage of time the source shows micro-variations compared to the total observation time. Romero [46] proposed an alternative formulation of the duty cycle calculated as follows:

$$DC = \left( \frac{\sum N\left(\frac{1}{\Delta t}\right)}{\sum \frac{1}{\Delta t}} \right), \tag{1}$$

where *N* is either 0 or 1 depending on whether micro-variability is observed or not, and $\Delta t$ is the duration of the observation corrected for cosmological effects. After testing the algorithm, we noticed that there is a bias for longer data sets that changes the duty cycle calculation significantly if many of the observations are different in length. Although we calculated the duty cycle using both methods, our discussion focuses on the more direct calculation based solely on time on versus time off.

The duty cycles for our objects are listed in Table 2. Column 1 lists the object name. Column 2 is the initial test result using the Howell method with the linear trend (if any) still present. Column 3 reports the duty cycle calculated with the Romano method, and Column 4 is our more direct method. Column 5 gives the results with the linear trend removed, while Columns 6 and 7 give the duty cycles calculated with both methods. Our sample exhibited an average OVV duty cycle of 36.21% (6 objects), while the average BL Lac is 33.24% (5 objects) with linear trends removed and using sources that had at least 50 data points. Note there are a few micro-variability curves that switched from "micro-variability" to "no micro-variability" as a result of the linear subtraction. It is interesting to note that four objects on our list showed no micro-variability, but we only observed those objects three times or less. The significance of the duty cycle to our model is the larger the duty cycle, the larger the Reynolds number is so that well-developed turbulence is more likely. Observationally, the larger the duty cycle, the more likely we will observe variations we can model. We should also note that the duty cycle calculation with only a few observations should be taken with perspective, such as 3C446 with only two observations, whereas S5 0716+71 and BL Lac have 77 and 31 observations, respectively. Obviously, the duty cycle calculations are unreliable with only a few observations; however, we wanted to report all of the data we acquired during our program.

### 4.1. Correlation with Flux Level

We studied the correlation between the occurrence of micro-variations and short and long-term variations in four blazars [4]. No correlation between the flux level and the presence of micro-variability is present in the data. The light curves in Figures A1–A15 show the monitoring observations, with black symbols. The dates when micro-variability observations were obtained are superimposed with red and green symbols. Inspection of the light curves presented in the Appendix A shows that we were successful in looking for micro-variability throughout their brightness ranges. We correlated the presence of micro-variability with brightness and calculated the point biserial correlation coefficient for each object. The correlation coefficient ($r_{pb}$) pertains to the case where one variable is dichotomous, and the other is non-dichotomous. By convention, the dichotomous variable is treated as the X variable; its two possible values are coded as X = 0 and X = 1, and the brightness is the non-dichotomous Y variable. We assigned X = 0 for no micro-variability and X = 1 when micro-variability was statistically present. The Y variable is assigned the average magnitude during the night the micro-variability curve was obtained. Column 6 of Table 1 lists the resulting correlation coefficients and the two-tailed probability distribution for each source. This calculation was performed for sources when there were more than five micro-variability observations. The largest correlation coefficient was +0.56 for 3C 454.3 with nine micro-variability observations, and the smallest was 0.02 for OJ 287 with 22 observations. The average bi-serial correlation coefficient for BL lacs was 0.25, while

the average for OVV quasars was 0.404. Thus, we conclude that there is no correlation between brightness and the presence/absence of micro-variability.

Howard [4] found a very weak dependence on the slope of the short-term variations over several days or months and the presence of micro-variability. Archival optical observations made during a two-month period surrounding micro-variability observations were used to calculate the slope of the short-term light curves for their four objects. Statistically, the micro-variations tend to occur more often when the slope of the short-term light curve is steep and much less often when it is flat. Thus, if this correlation is real, during the course of a short-term outburst, one would not expect to see micro-variability at the maximum or minimum flux levels but during the rise and declining phases of the outburst. We were unable to perform further tests of this idea with our data because of the lack of monitoring data around the micro-variability observations.

### 4.2. Theory of Shock Waves Encountering Turbulent Cells

The results reported in the previous sections left us with a sense that non-periodic intermittent pulses make up micro-variations. The detection of pulses from individual turbulent cells might give us information about the plasma obtainable in no other way. After studying hundreds of micro-variability light curves published in the literature and observations acquired because of this program, we can summarize the results of various types of analyses as follows:

1. The occurrence of micro-variations showed no correlation with flux levels;
2. Micro-variability is intermittent in each source;
3. No consistent underlying periods were found in the micro-variations from one light curve to the next for any object;
4. The Power Spectral Density (PSD) of the individual light curves did not consistently show a pure noise spectrum [8,9].

The micro-variability light curves can be interpreted as being composed of individual pulses or shots rather than random noise or full-wave periodic oscillations. Studies of the rise time and decay time in the micro-variations yielded consistent values for the slopes in many different light curves and many different sources [15,47], indicating some structure is present. We focused on the micro-variations that pulsed and began to look for possible sources of such pulses in the context of a relativistic jet model. If a shock encounters a turbulent cell, it accelerates electrons in the cell, and the electrons cool by synchrotron emission. The synchrotron emission depends on the local magnetic field direction, strength, and electron density. Lehto [48] first investigated shot models to describe the apparent $1/f$ power spectra seen in X-ray light curves of active galaxies. They assumed that the shots had a delta function rise and exponential decline and were able to show that a random distribution of such shots yields a $1/f$ power spectrum. Lehto's pulses did not resemble the micro-variability pulses seen in S5 0716+71 or in ON 231, but the idea that the micro-variations could be deconvolved into individual shots seemed compelling. Visual inspection of over 100 micro-variability light curves between 1989 and 2009, from this program and from [5], led [7] to propose that the micro-variability curves were indeed shots, but shot profiles similar to the profiles derived by [49]. Our model is based on individual synchrotron turbulent cells energized by a plane shock propagating down the jet [7]. Turbulence in the jet forms individual cells with a range of sizes and magnetic field strengths and orientations. The micro-variations observed in S5 0716+71, ON 231, and other objects are then be interpreted as a convolution of individual synchrotron pulses in the turbulent jets.

## 5. Theoretical Model

### 5.1. Theory of Shock Waves Encountering Turbulent Cells

The theory of a shock wave encountering a cylindrical plasma density enhancement was worked out in [49] (hereafter KRM). KRM calculated the particle acceleration in the shock front for various magnetic field orientations and particle densities. They present dif-

ferent profiles for different cases where the acceleration and cooling times can be calculated, yielding constraints on the cooling length L, the magnetic field strength B, and orientation θ. The profiles described by KRM look similar to the micro-variability pulse profiles seen in the actual micro-variability curves. Thus, it seems compelling to re-examine this basic KRM model in terms of the situation where the shock front encounters multiple turbulent cells instead of a single cylindrical density enhancement. First, we recalculated the KRM burst profiles, and then we could compare these profiles to our micro-variability curves. Figure 1 shows the general model attributes with the jet consisting of turbulent cells. The particle distribution function is given by [50] as:

$$\frac{\partial N}{\partial t} + \frac{\partial}{\partial \gamma}\left[\left(\frac{\gamma}{t_{acc}} - \beta_s \lambda^2\right) N\right] + \frac{N}{t_{esc}} = Q\delta(\gamma - \gamma_0) \tag{2}$$

where

$$\beta_s = \frac{4}{3}\frac{\sigma_t}{m_e c}\left(\frac{B^2}{2\mu_0}\right). \tag{3}$$

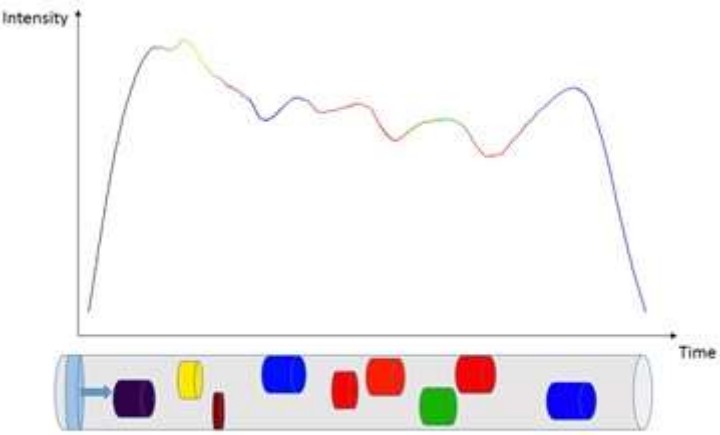

**Figure 1.** Diagram showing the general model attributes with the jet consisting of turbulent cells.

The quantities in the above Equations are $N$ = number of particles as a function of time ($t$) and energy ($\gamma$). $t_{acc}$ is the acceleration time, $\beta_s$ is the synchrotron emission, and $t_{esc}$ is the timescale for particle escape. KRM solves this Equation for the case where there is a constant injection rate $Q_0$ after switch-on at t = 0. The solution is of the form:

$$N(\gamma, t) = a\frac{1}{\gamma^2}\left(\frac{1}{\gamma} - \frac{1}{\gamma_{max}}\right)^{(t_{acc} - t_{esc})}\Theta(\gamma - \gamma_0)\Theta(\gamma_t - \gamma) \tag{4}$$

where the source term is:

$$P(\nu, \gamma) = \sqrt{3}e^2\Omega\left(\frac{sin\theta}{2\pi c}\right)z^{0.3}\exp(-z) \tag{5}$$

and

$$z \equiv \frac{4\pi\nu}{3\Omega \sin\theta\delta^2} \tag{6}$$

The variables $t_{acc}$, $t_{esc}$ B, and $\theta$ are physical parameters that determine the pulse rise and decay time and amplitude. This calculation is performed for each cell in the plasma. The magnetic field B and the angle of the field relative to the line of sight $\theta$ are assumed to be different for each cell in a turbulent medium. Thus if a micro-variation is light from a single cell, the magnetic field could have an arbitrary orientation relative to the t4e line of sight. Thus, if we can observe polarization percentage and angle with a time resolution of resolving individual pulses, we would expect to see random swings in polarization angle

and percentage. Most polarization observations do not have the time resolution to pick out individual cells (see Section 5.4).

### 5.2. Fitting Pulse Profiles

The solution of Equations (2) through (6) gives the amplitude and shape of an individual pulse representing the emission from a single turbulent cell. The micro-variability light curve would then be a convolution of many cells being energized as the shock propagates through the three-dimensional jet medium, encountering a number of cells. Our one-dimensional time series cannot specify where in the 3-D jet the individual cells are located, just the arrival times of the pulses. Future work is envisioned to take 3-D simulated jet turbulence and run a simulated shock wave through it to observe whether the light curves we obtain from it look like our micro-variability curves (50). We can reduce the problem in first-order to a linear combination of many pulses convolved together:

$$I(\nu, t) = I_{laminar} + \sum I_{cell(\nu,t)} \tag{7}$$

The above solutions determine pulse shapes as a function of frequency for a given amplitude and cell size. Figure 2 shows sample pulse shapes for frequencies corresponding to the center frequencies of the V and I optical bands superimposed in the left plot and the spectral index evolution in the right-hand plot. These pulse shapes can then be fitted to actual light curves to determine if the pulse shapes fit the actual micro-variability curves well.

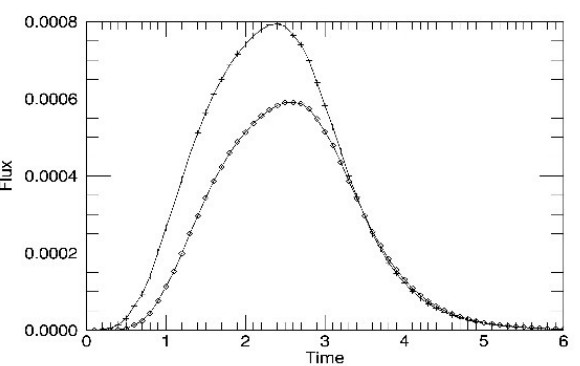 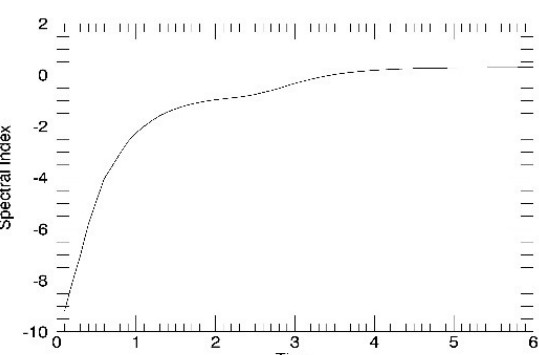

**Figure 2.** Theoretical pulse profiles for the V and I optical bands are presented in the **left** panel. The time axis and flux axes are in arbitrary units. The evolution of the spectral index is plotted in the **right** panel showing the expected evolution of the spectral index during a single pulse at these frequencies.

The micro-variability curves are currently fit by identifying the number of distinct, well-sampled peaks and determining the amplitude above the linear background. We then determined the Full Width Half at Maximum (FWHM) and a center time for each pulse. The FWHM, center time, and amplitudes are adjusted until we can obtain a goodness of fit greater than 95%. This was performed for a number of micro-variability curves, including ON 231 and S5 0716+71. Ideally, one could always deconvolve micro-variability curves into their component pulses, but in practice, some micro-variability curves may be too complex with many smaller unresolved pulses and the data too sparse to accurately deconvolve the individual pulse profiles. An example of a light curve where the pulses can be deconvolved with reasonable accuracy is shown in Figure 3. The micro-variability curve of ON 231 is fit with seven independent pulses. The resulting correlation coefficient of 0.995.

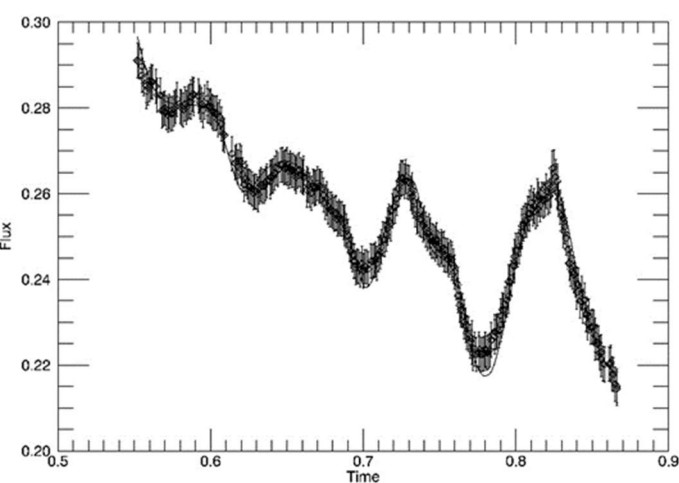

**Figure 3.** ON 231 with 7 pulses convolved together.

By fitting a number of pulses in many micro-variability curves, we can obtain an idea of the cell size distribution (assuming a shock speed). These micro-variability pulses are superimposed onto a background magnetic field that could be twisted as a function of distance from the core, as we further discuss below. The size scale of the turbulent eddies is important information for people who model relativistic turbulence in jets of this type. The distribution of pulse sizes would be of interest to turbulence specialists who model these systems. Zrake [51] showed that as the turbulence grows, the magnetic cells become larger. Table 3 shows the sources and data sets for which we have estimated cell sizes by fitting the pulses with our model.

**Table 3.** Cell Size Results.

| Object | Observation | # Of Cells | Average Cell Size (AU) | Minimum Cell Size (AU) | Maximum Cell Size (AU) | Reference |
| --- | --- | --- | --- | --- | --- | --- |
| 0716+71 | WEBT | 35 | 36.5 | 6.5 | 164 | [15] |
| Zw 229-15 | Kepler 6 and 7 | 80 | 37.34 | 10 | 120 | [52] |
| KA 1925+50 | Kepler 6 and 7 | 107 | 35 | 10 | 100 | [52] |
| KA 1858+48 | Kepler 6 | 46 | 37 | 10 | 130 | [52] |
| KA 1904+37 | Kepler cadences 6 and 7 | 25 | 37 | 20 | 130 | [52] |
| 0716_71 | 37 individual micro-variability observations | 231 pulses | 12.8 | 4.9 | 79.1 | [53] |
| 0716+71 | Over 8 years of micro-variability light curves | | 20 | 5 | 55 | [54] |

It should be noted that the cell sizes are based on the choice of shock speed. Therefore, the actual value is highly dependent on the choice of this speed, which is not directly observable. More important is the distribution of cell sizes that is consistent with turbulent cell size distributions seen in simulations [55].

### 5.3. Testing the Model

The pulse shape fits available micro-variability curves well, but we can apply a further test. A more sophisticated test of the model is to take the multi-wavelength micro-variability fits and calculate the spectral indices for each individual pulse. The equations to calculate pulse shapes are frequency-dependent. Color effects offer a test as to the validity of the model. Clements (46) investigated color changes during micro-variations and found that BL Lacertae became slightly bluer as it increased in flux during the micro-variations. In order to test this prediction, multi-color micro-variability observations were made, sometimes alternating filters on a single telescope, other times using SARA North and SARA South simultaneously but with different filters. These observations yielded simultaneous V and R micro-variability curves in two or more bands. Figure 4 is an example of this type of observation. Every data set we tested with this method produced results consistent with our model [54,56,57]. In Figure 4, we fit four pulses, but the first one was incomplete, and the second was very low amplitude. The third and fourth pulses are well defined enough to test the model. The fit has a correlation coefficient of 0.94 and 32 degrees of freedom.

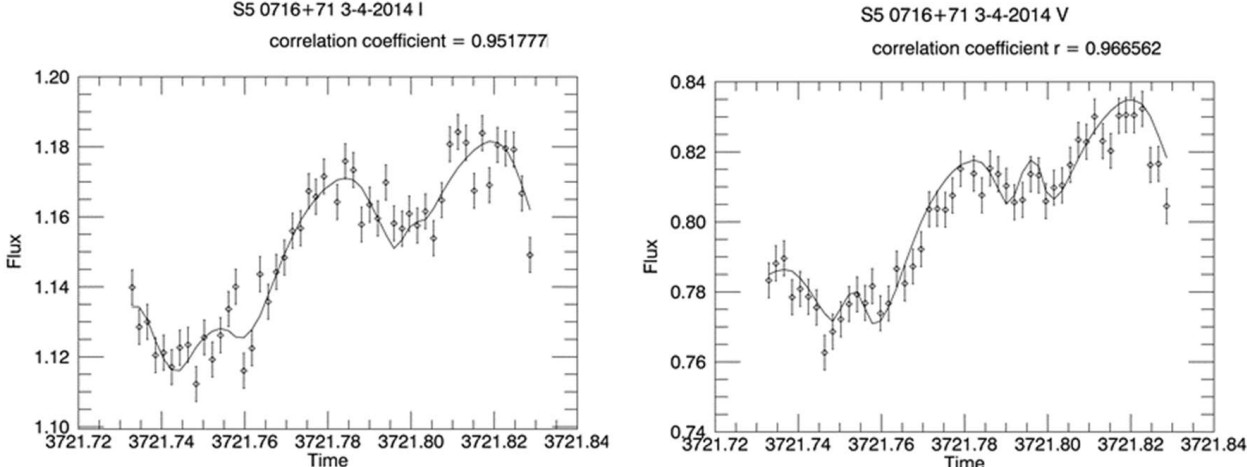

**Figure 4.** Micro-variability light curve for S5 0716+71 from 3 April 2014 fit with 4 pulses centered on the V and I bands. The frequency bands were fit independently of each other, and both yielded a goodness of fit exceeding 0.94 with DOF = 32.

Figure 5 shows the model pulses derived from the fitted light curve for pulse 3 in 0716+71 on 3 April 2014. This result agrees with the predicted spectral behavior, which is shown in Figure 2. The evolution of the spectral index during the pulse also agrees with the predictions of the model (Figure 2). Each data set we applied this test with yielded similar results [54]. Xu [54] used our model and developed an automated fitting program to fit light curves of S5 0716+71 taken in the VRI bands from 2011 to 2018. Every time we made high S/N multi-frequency fits, the results were consistent with the model's multi-frequency predictions.

We fit 37 published micro-variability light curves from [5] with our model, achieving a high degree of correlation and goodness of fit of 0.99 in each micro-variability curve [58]. We also used our model to fit a variety of other data sets, including Kepler [52,59] observations and TESS observations [58]. The Kepler and TESS observations were not made in the UBVRI system we normally use, so we solved the pulse shape equations for the center frequency of the Kepler and TESS detectors and fit those pulses to the data. Dhalla analyzed the Kepler light curves of four Kepler Quasars: Zw 229-15, KA 1925+50, KA 1858+48, and KA 1904+37 [52,59]. She obtained a goodness of fit of 0.99 for six of the seven data sets with over 4100 degrees of freedom. She fit, on average, 25 to 55 pulses to each observation with average cell sizes ranging from 28 AU to 54 AU, given a shock speed of 0.9c. Fitted cells showed a range of sizes as expected from a turbulent plasma with many more smaller

cells than larger ones in a nearly linear distribution. There are, of course, many more cells present than we can see, but the largest ones are represented in the micro-variability curve.

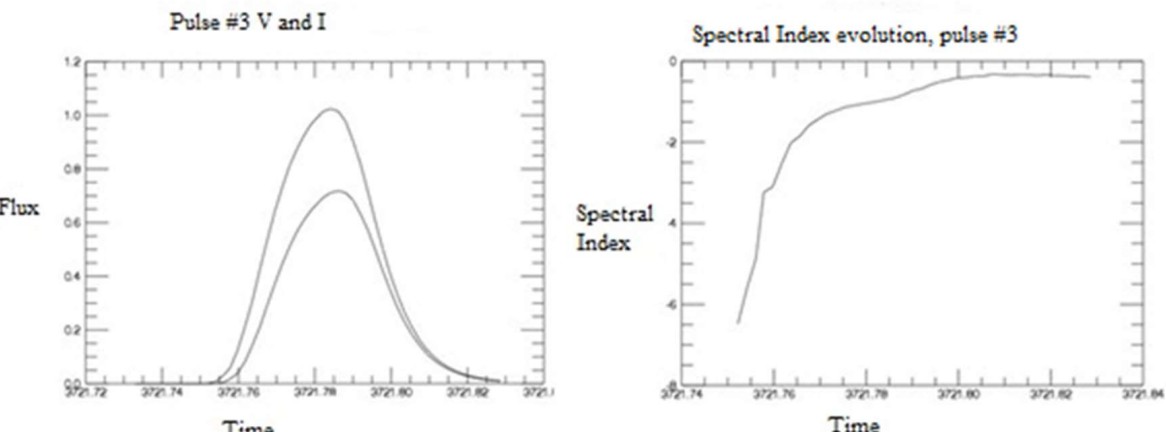

**Figure 5.** Pulse 3 for the S5 0716+71 observation from 3 April 2014. Pulse 3 was fit independently with a model pulse for V and I bands. This pulse was then superimposed (left panel), and the spectral index evolution was calculated (right panel). These observations were made with the SARA 0.9-meter telescope at Kitt Peak National Observatory.

The TESS micro-variability light curve of BL LAC from [58] is shown in Figure 6 with our model superimposed. The fit yielded a correlation coefficient of 0.97973 with 13 pulses. The range of cell sizes was calculated assuming a shock speed of 0.9c and a Doppler factor of 7. The resulting cell sizes spanned the range 44 AU < cell size < 147 AU. The model fit yielded a correlation coefficient of 0.97 with 98 degrees of freedom. More work on this light curve is in progress. Weaver [58] concludes that the TESS light curve variation and the WEBT polarization observations are naturally explained in terms of an underlying magnetized turbulent jet with a large number of turbulent cells. We fit the TESS data with our model (Figure 6).

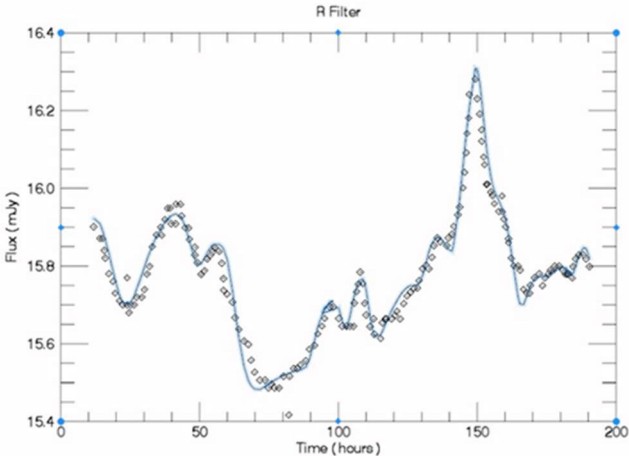

**Figure 6.** The TESS light curve of BL Lac. Correlation coefficient of r = 0.979.

*5.4. Polarization Predictions and Test*

Polarization is another important test of this model. Each turbulent cell should have its own randomly oriented magnetic field. The amplitude of the pulse depends on the density of the cell, the magnetic field direction, and intensity. Each pulse should correspond to an individual turbulent cell. We attempted to test this prediction with the CIRCE IR detector on the 10.4- meter telescope. We observed object S5 0716+71 because of its high duty cycle for micro-variability [60]. We observed the H band with polarimetry. Both

polarization direction and degree were measured at a time resolution of about 1 s in that band. Simultaneous VRI observations were obtained with the SARA JKT also on La Palma. Unfortunately, the object failed to show any micro-variations, so we were unable to test the model completely. However, it also failed to show swings in the polarization direction and percentage, consistent with our model. Since [58] also published the polarimetry from the WEBT along with the Tess light curve, we attempted to test this aspect of the model by comparing the fitted cells with the polarization variations published. The published polarization data were not dense enough to separate out individual pulses, but this remains a possible direct test of the model.

Sasada [61] made photo-polarimetric observations of S5 0716+71 in the V and NIR bands, measuring both the flux and the Q and U Stokes parameters with a time resolution of about 5 min. They noted a "bump" (micro-variation) in the light curve that corresponded to a change in the polarization by about $27 \pm 5\%$. They concluded that "the emitting region of the micro-variability in the optical band is a small and local area compared with the whole optical emitting region in the jet" [61]. This is consistent with our model predictions for polarization.

## 6. Conclusions

We presented systematic observations in a sample of blazars and presented a physical explanation of the observed micro-variability seen in those observations. We used statistical methods to verify the existence of micro-variability in the data sets and propose a new definition of micro-variability that differentiates short-term linear trends from micro-variability. We define micro-variability as oscillations around any linear increase or decrease in flux level. The linear increase or decrease may be due to a background twist in the magnetic field of the jet. We calculated duty cycles for each object for which we had sufficient observations. Based on these data, we propose that micro-variability is the result of a shock encountering a turbulent region of the jet flow. If the jet flow is laminar or the turbulence is not well developed in the region of the shock, then we do not see any micro-variability. The pulses from larger individual cells encountering the shock wave are convolved together, and we see them as micro-variations. We can only identify individual cells that are larger than our sampling rate and photometric accuracy. In addition to the larger pulses we can actually fit, there are many smaller cells that we cannot detect. Many authors conclude that the polarization changes and swings are due to turbulence in the jet flow, but they do not associate individual peaks in the micro-variability light curve with resolvable pulses from individual cells of resolvable size [58,62,63]. Our model offers a natural explanation for the erratic swings in polarization direction and degree seen in these sources. Unless you can obtain a polarization observation of an individual pulse, you would obtain a convolution of the polarization of many pulses, randomly oriented. This effect will produce random and rapid polarization swings such as the ones observed [61,62].

The size distribution of cells derived from the 7-h WEBT observation of 0716+71 [15] showed the same distribution that would be expected from a turbulent plasma. We show the distribution of cell sizes determined from thirty-seven individual micro-variability observations of S5 0716+71. The analysis yielded 231 pulses, which are plotted in Figure 7. Figure 7 shows that the distribution of cell sizes is consistent with the Kolmogorov distribution and consistent with the distributions seen in relativistic turbulence simulations $(k \sim Ak^{-5/3})$ [12].

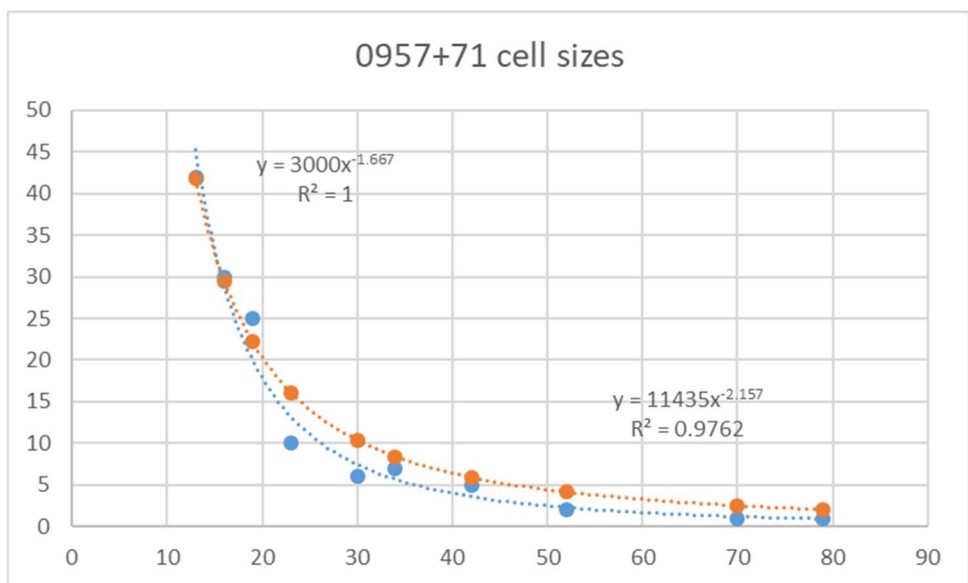

**Figure 7.** The blue points are the cell sizes as derived fitting pulses to 231 micro-variability curves of 0957+56 [53]. The red curve is the Kolmogorov distribution $k^{-5/3}$.

The value of associating the micro-variability pulses with individual turbulent cells is that if we reasonably estimate the shock speed, we can obtain an idea of the actual cell sizes by the width of the pulses. The amplitude of the pulse over the background flux gives us an idea of the density enhancement convolved with the magnetic field intensity and direction. Magnetic strength and orientation can be deconvolved with polarization observations. These results can be used to inform theoretical simulations of relativistic plasmas.

Thus, the model has passed every test that has been applied to it so far: pulse fitting, spectral index changes, polarization changes, and even yields cell size distribution indicative of a turbulent plasma.

**Author Contributions:** J.R.W. is responsible for initiating this research, supervising the observations and data reduction, and writing the final paper. V.A.: CCD reduction of data after 2019 and updating all statistics. All other authors contributed by making observations and data reduction, and model construction. All authors have read and agreed to the published version of the manuscript.

**Funding:** This work was supported in part by NSF grant 0324238. GB acknowledges the financial support by Narodowe Centrum Nauki (NCN) grant UMO-2017/26/D/ST9/01178.

**Institutional Review Board Statement:** Not applicable.

**Informed Consent Statement:** Not applicable.

**Data Availability Statement:** The data is available via the first author.

**Acknowledgments:** The authors would like to thank the anonymous referees for suggestions that improved the presentation. Based on observations obtained with the SARA Observatory 0.9 m telescope at Kitt Peak, which is owned and operated by the Southeastern Association for Research in Astronomy (saraobservatory.org). The authors are honored to be permitted to conduct astronomical research on Iolkam Du'ag (Kitt Peak), a mountain with particular significance to the Tohono O'odham Nation.

**Conflicts of Interest:** The authors declare no conflict of interest.

## Appendix A

Light curves of the sources in our study.

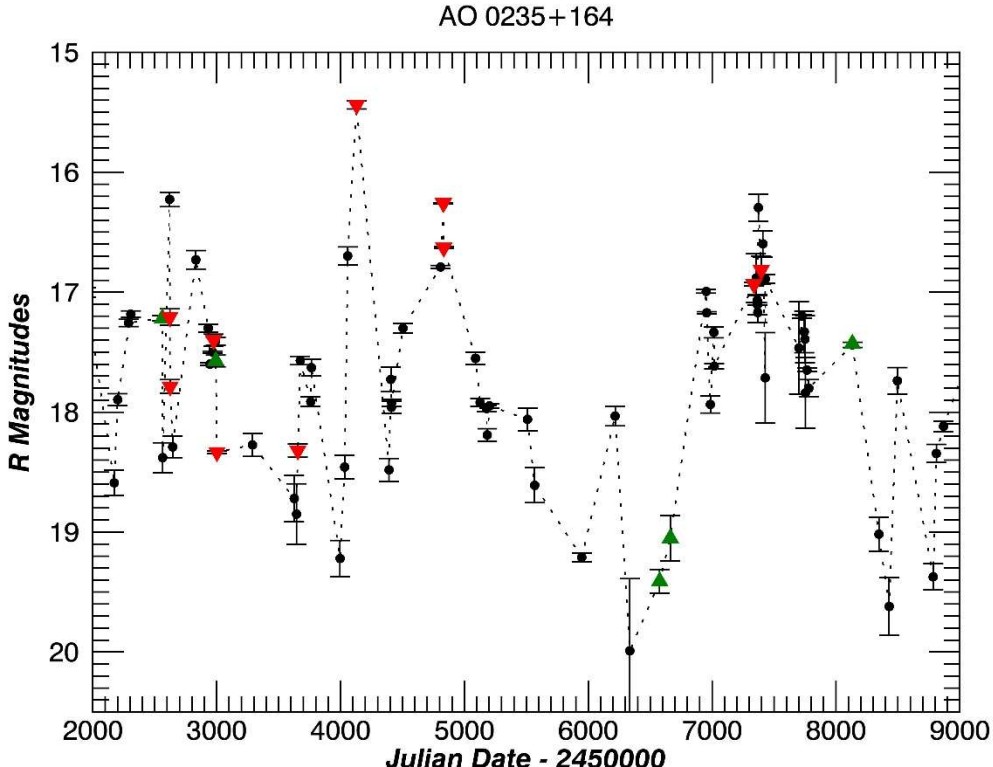

**Figure A1.** The light curve of AO 0235+164. The black symbols are monitoring observations, the red symbols indicate micro-variability was not seen, and the green up-pointing triangle symbols indicate when micro-variability was seen.

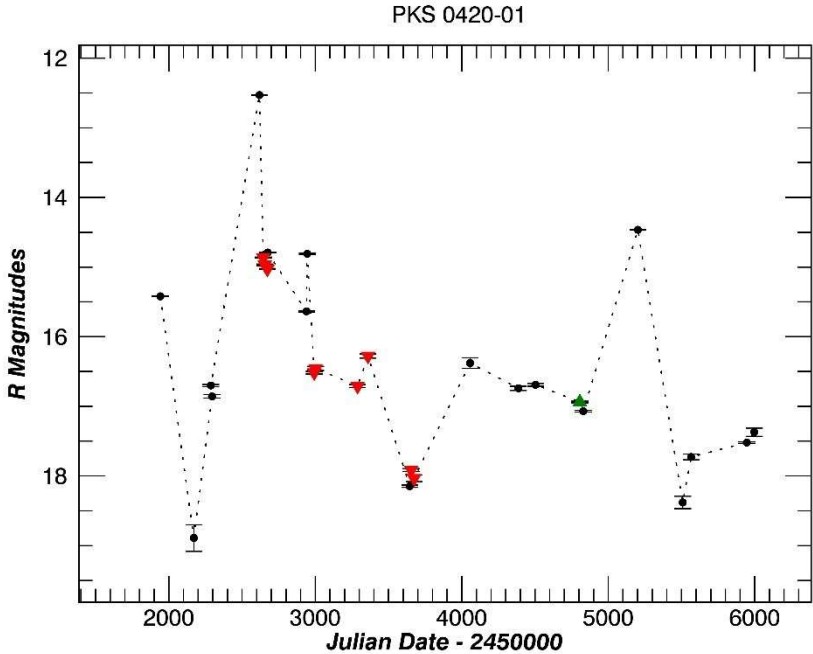

**Figure A2.** The light curve of PKS 0420-01. The black symbols are monitoring observations, the red symbols indicate micro-variability was not seen, and the green up-pointing triangle symbols indicate when micro-variability was seen.

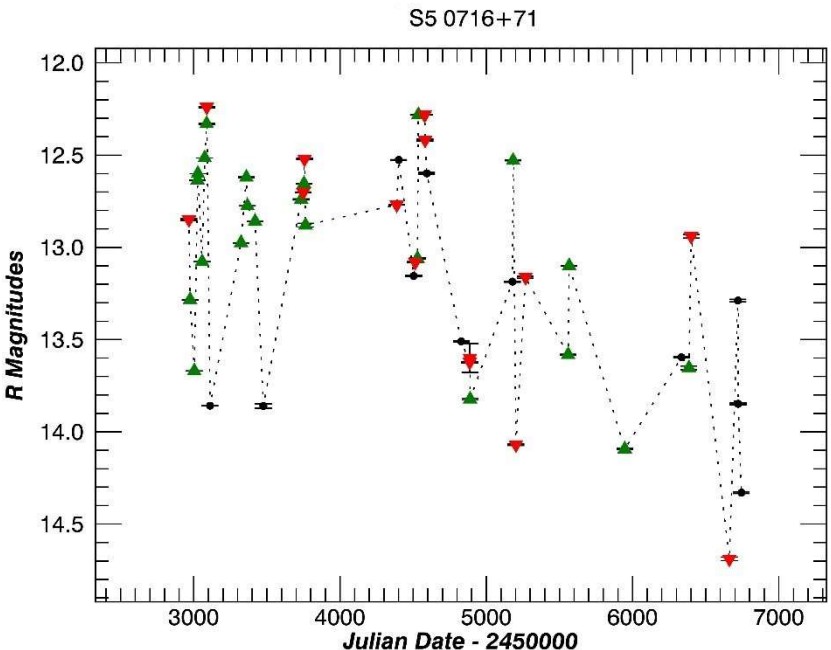

**Figure A3.** The light curve of S5 0716+01. The black symbols are monitoring observations, the red symbols indicate micro-variability was not seen, and the green up-pointing triangle symbols indicate when micro-variability was seen.

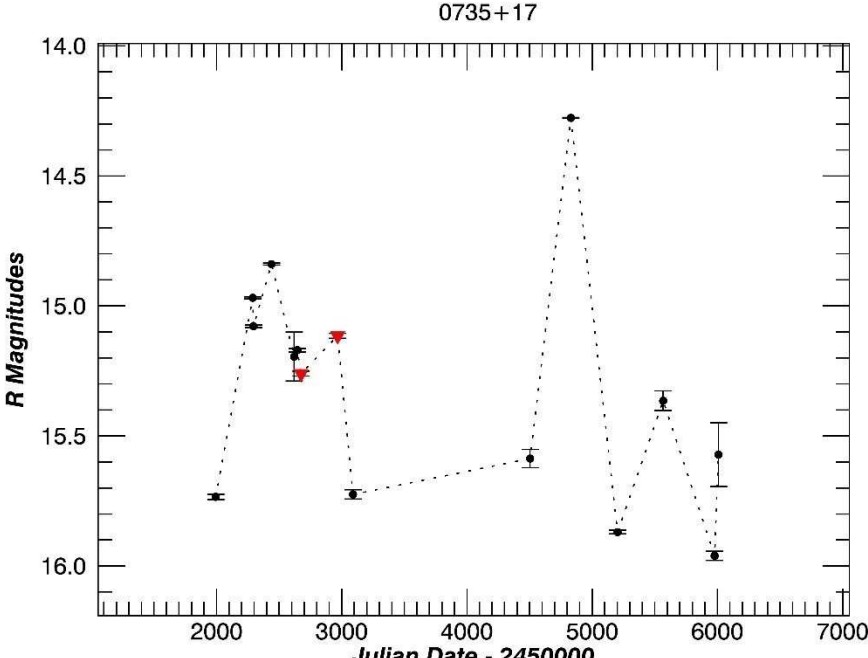

**Figure A4.** The light curve of PKS 0735+17. The black symbols are monitoring observations, the red symbols indicate micro-variability was not seen, and the green up-pointing triangle symbols indicate when micro-variability was seen.

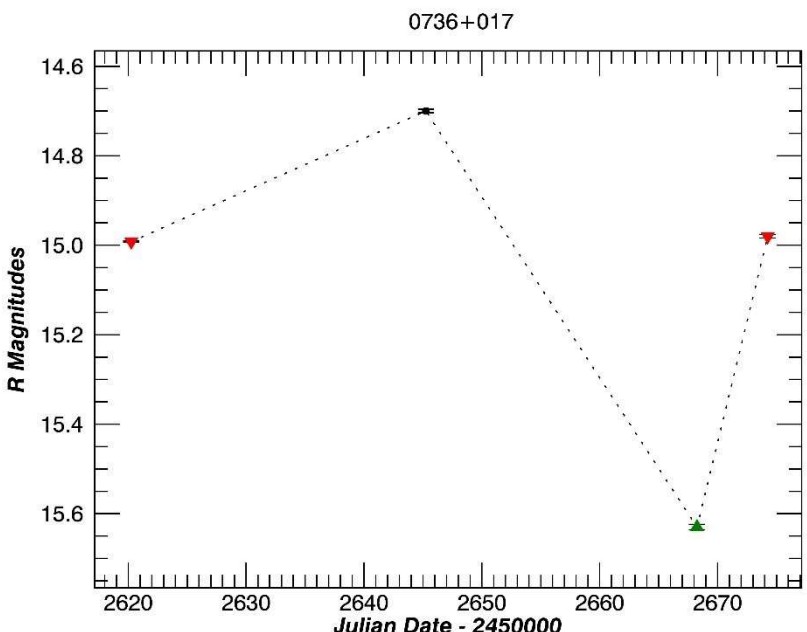

**Figure A5.** The light curve of PKS 0716+01. The black symbols are monitoring observations, the red symbols indicate micro-variability was not seen, and the green up-pointing triangle symbols indicate when micro-variability was seen.

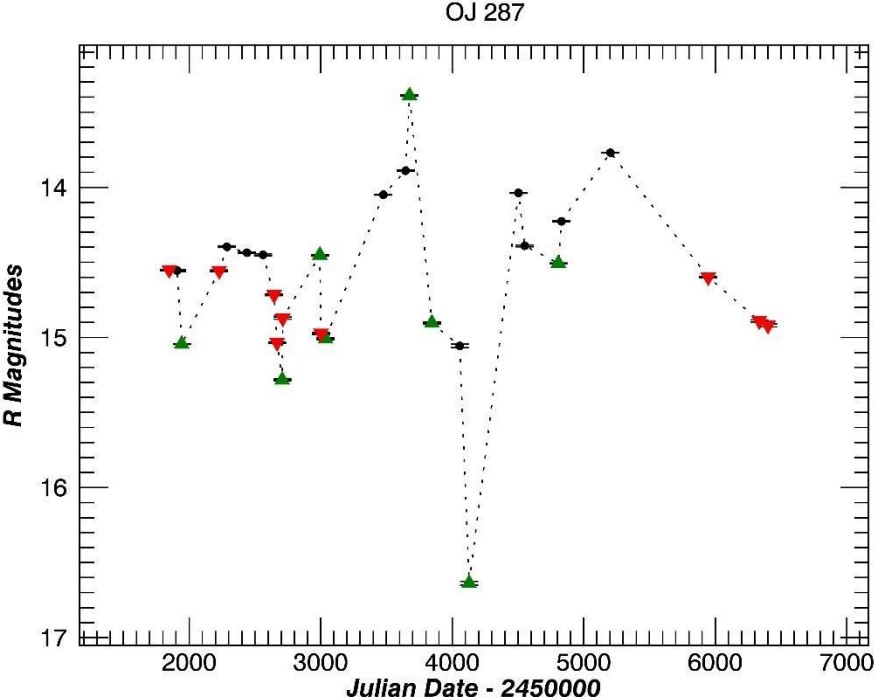

**Figure A6.** The light curve of OJ 287. The black symbols are monitoring observations, the red symbols indicate micro-variability was not seen, and the green up-pointing triangle symbols indicate when micro-variability was seen.

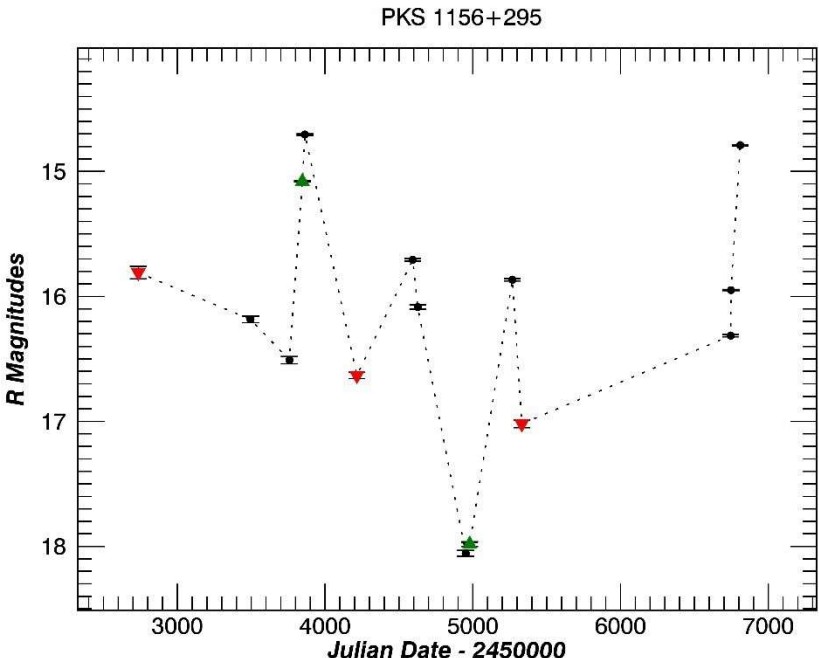

**Figure A7.** The light curve of PKS 1156+295. The black symbols are monitoring observations, the red symbols indicate micro-variability was not seen, and the green up-pointing triangle symbols indicate when micro-variability was seen.

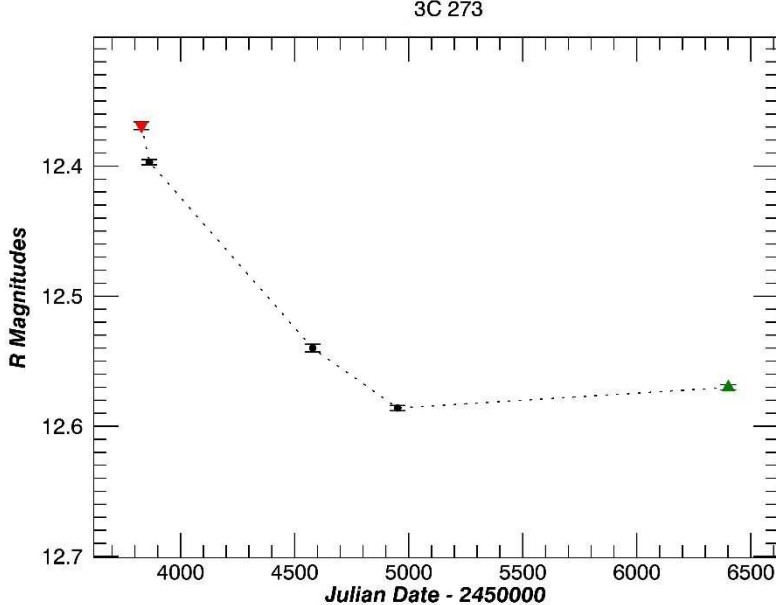

**Figure A8.** The light curve of 3C 273. The black symbols are monitoring observations, the red symbols indicate micro-variability was not seen, and the green up-pointing triangle symbols indicate when micro-variability was seen.

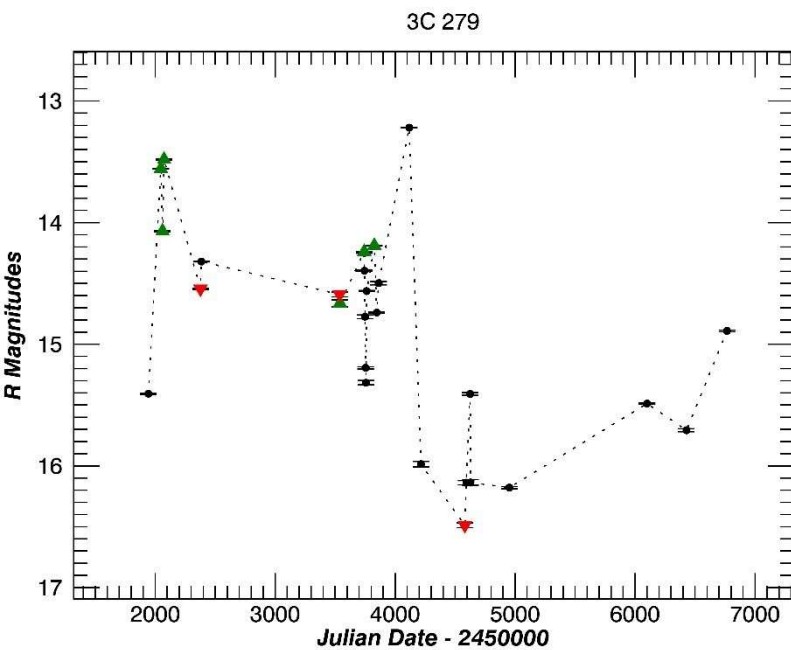

**Figure A9.** The light curve of3C 279. The black symbols are monitoring observations, the red symbols indicate micro-variability was not seen, and the green up-pointing triangle symbols indicate when micro-variability was seen.

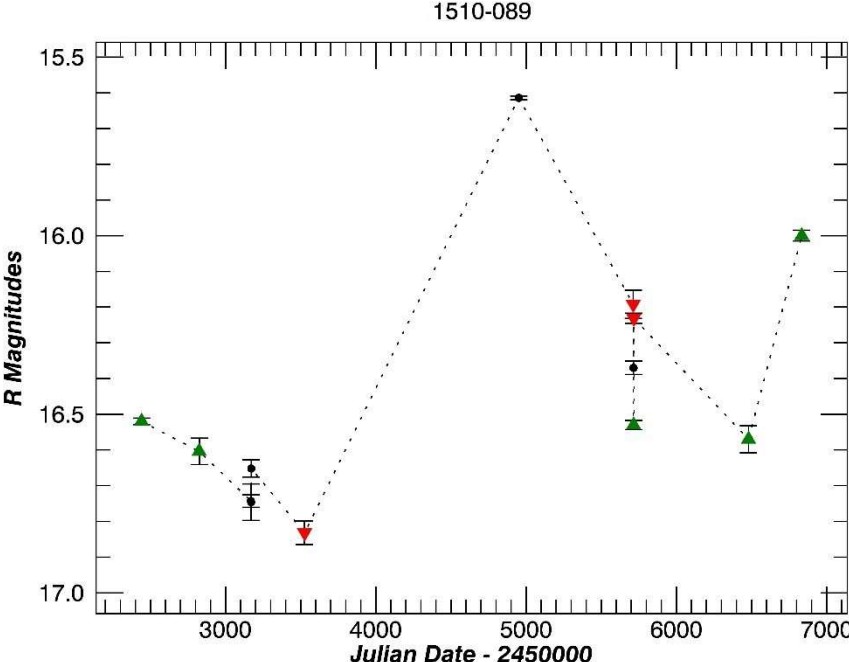

**Figure A10.** The light curve of 1510-089. The black symbols are monitoring observations, the red symbols indicate micro-variability was not seen, and the green up-pointing triangle symbols indicate when micro-variability was seen.

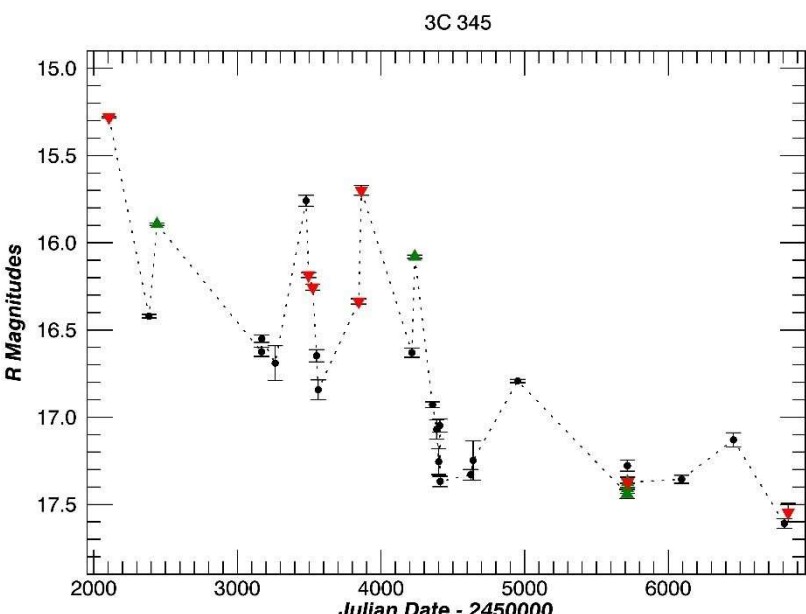

**Figure A11.** The light curve of3C 345. The black symbols are monitoring observations, the red symbols indicate micro-variability was not seen, and the green up-pointing triangle symbols indicate when micro-variability was seen.

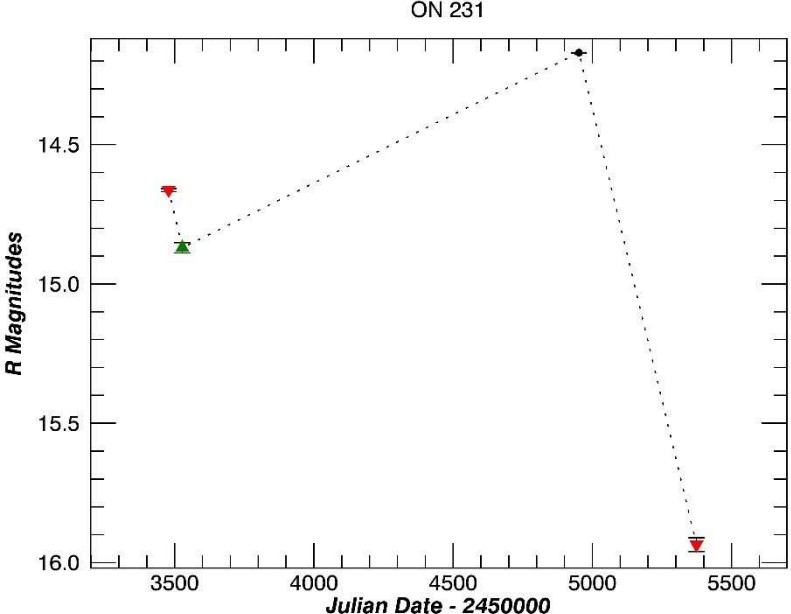

**Figure A12.** The light curve of ON 231. The black symbols are monitoring observations, the red symbols indicate micro-variability was not seen, and the green up-pointing triangle symbols indicate when micro-variability was seen.

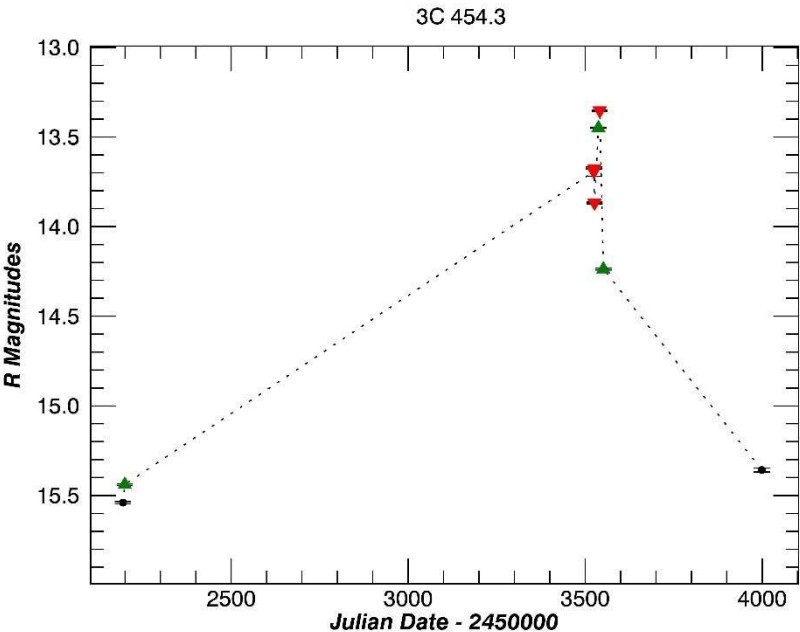

**Figure A13.** The light curve of 3C 454.3. The black symbols are monitoring observations, the red symbols indicate micro-variability was not seen, and the green up-pointing triangle symbols indicate when micro-variability was seen.

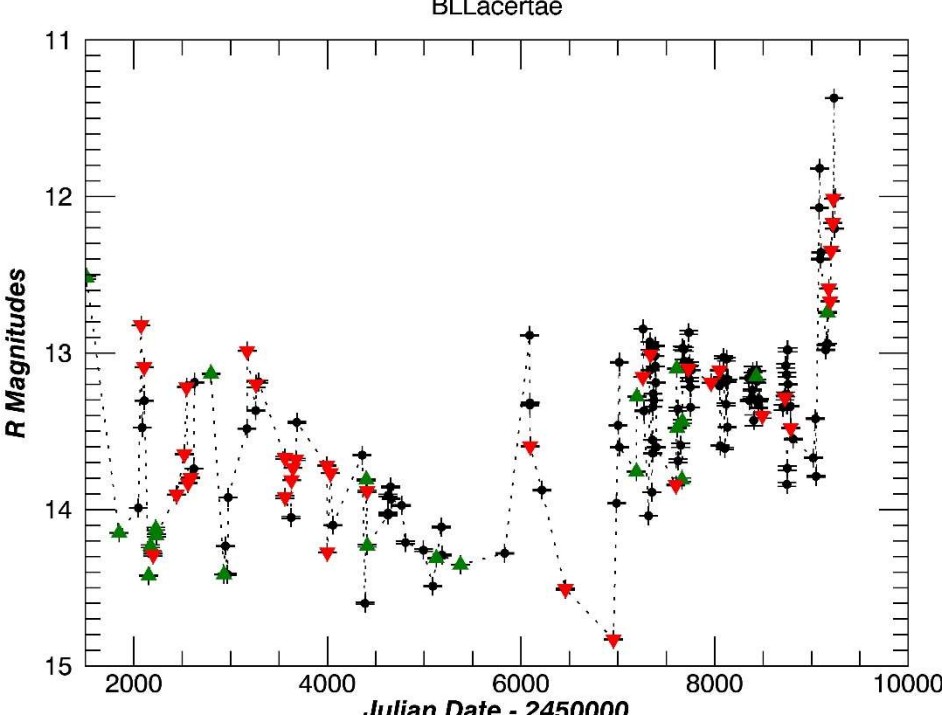

**Figure A14.** The light curve of BL Lac. The black symbols are monitoring observations, the red symbols indicate micro-variability was not seen, and the green up-pointing triangle symbols indicate when micro-variability was seen.

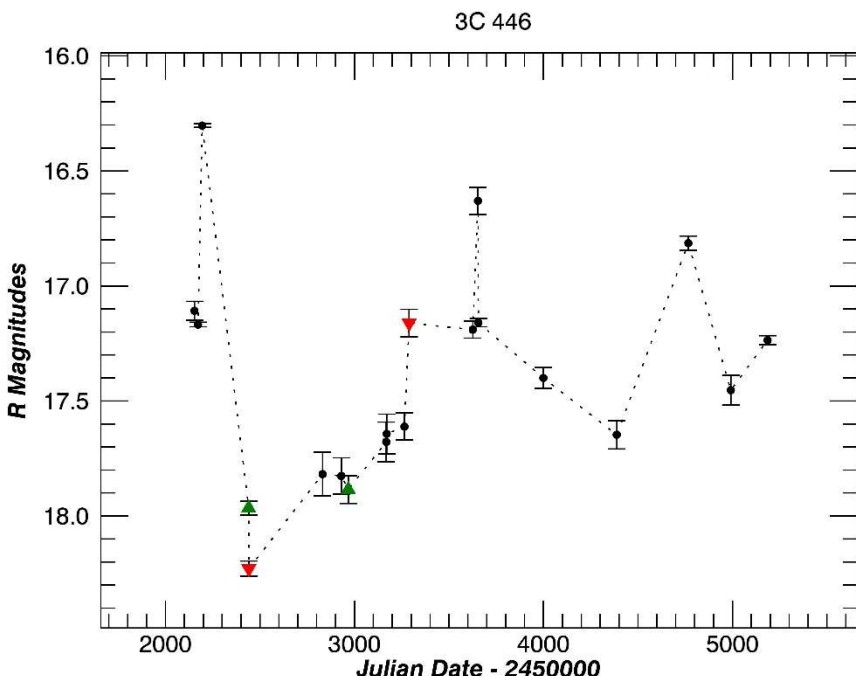

**Figure A15.** The light curve of 3C 446. The black symbols are monitoring observations, the red symbols indicate micro-variability was not seen, and the green up-pointing triangle symbols indicate when micro-variability was seen.

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
