# Peer review of "The Nature of Micro-Variability in Blazars"

_galaxies, doi:10.3390/galaxies9040114_

Round 1

Reviewer 1 Report

I have reviewed the manuscript "The Nature of Micro-variability in Blazars". While the results are certainly interesting there are several issues, mostly with the presentation of the analysis and results that prevent me from recommending it for publication in the present form. I list my comments below.

Section 2, Materials and Methods: This section provides minimal information. Instead the authors have spread relevant information throughout the text causing confusion. For example, they discuss the duty cycle in section 3 but they define it in section 4 which is supposed to be the Discussion section. I suggest the authors restructure the paper in a way that their analysis steps are clear. 

Section 3, Results: Here the authors start discussing the individual sources of their sample. This is hardly the place to do that. I would suggest the authors either create a new section titled "Sample" or something similar, or move that information to an appendix. In addition, there are bits and pieces of irrelevant information. For example, while the theme of the paper is microvariability, the authors start discussing periodicities of 5.7 years in AO 0235+164 or for PKS 0420-01 the state "We analyzed the micro-variations in a number of sources including PKS 0420-01, PKS 0736+01, PKS 1510-08, 3C 273, ON 231 and BL Lac [8]. They reported periods of 2.4 hours and 1.413 hours in micro-variability curves for 0420-01 by using unequal interval FT analysis." which is a bit peculiar. I suggest the authors trim the information they provide only to essentials and only for the specific source they are discussing.

Section 4, Discussion: subsection 4.1, 4.2, 4.3, 4.4 present analysis and modeling that is not appropriate for a discussion section. Those belong in section 3

section 4.2: There is hardly any explanation of the model or what the different variables correspond to.

The authors claim to have found microvariability in a number of light curves from their sample, however, they only apply the model to sources observed by space missions and 0716+71. If their model is not applied to their dataset, either the modeling is an irrelevant part of the paper, or the SARA sample is.

It would be helpful if the authors presented also one light curve of a source that has a linear trend and one that does not show either linear trends or microvariability for comparison.

Very few details are given for the "long term variability" the authors claim to study apart from a correlation analysis that lacks details. 

Minor comments:

Line 13 (and throughout the text): OVV is an outdated acronym that is not used any more. The frequently used term nowadays is Flat Spectrum Radio Quasar (FSRQ)

Line 25: "is an important" --> "is important"

Line 36: BL Lac objects are a subclass of blazars

Line 61 (and other places in the text): Kolomogorov --> Kolmogorov

Line 96:  a reference or a footnote with the website of the "Heidelberg Blazar web site" would be helpful

Table 1: the classification is not consistent, e.g., OVV quasar, QSO, and OVV are all FSRQs. Also the ordering of the table is not correct towards the end (I assumed it was done by increasing RA)

Line 314: DC is not defined 

Table 2 is hard to read. I suggest the authors have three columns regarding the number of light curves. One column with the number of light curves that do not show neither microvariability nor linear trends, a second column with the number of light curves that show linear trends but not microvariability, and a third column that shows the number of light curves with microvariability.

Line 416: emiision --> emission

Line 418; x-ray --> X-ray

Figures: The figures of the paper are low quality and at least some of them appear to be screenshots. The authors should provide better quality figures.

Line 559: " CIRCE IR detector on the 10.4- meter telescope." is repeated twice.

Author Response

I have uploaded a file with my responses.

Reviewer 2 Report

====
General: 

The manuscript presents a comprehensive optical photometry data set on 15 blazars, including novel, dedicated micro-variability observations. The duty cycle of the micro-variability and the correlation between the presence of micro-variations and the brightness of the source are investigated. A turbulent jet model that can explain the observed micro-variability is proposed. The scientific topic of blazar micro-variability is of high interest, as the time scale is very short for the system and hence providing an opportunity to probe the structure of the jet. 

The micro-variability observing strategy to cover a large range of brightness is in principle appropriate for the investigation. However, in practice, the small number of micro-variability observations in some of the sources not only limits the statistical power to investigate the bi-serial correlation between micro-variability and brightness, but also introduces potentially large bias in the estimation of the duty cycle. E.g., for the source PKS 0736+017, duty cycles were given based on three observations, and the duty cycle was estimated to be 0 for a few sources based on two or three observations. These results obtained from a small number of observations are questionable at best, and do not provide useful information. The statistical methods for the estimation of duty cycle, and potentially bi-serial correlation, should be improved. The authors could consider using Monte Carlo methods to test hypothetical duty cycles from 0 to 100% (in sufficiently small discrete steps) against the observing strategy and the results and give the confidence interval of the duty cycle. It is also necessary and important to explain the criteria/conditions based on which the decision of taking a micro-variability observation was made. 

It is unclear what conclusions can be drawn from the duty cycle results. 

Polarization results were mentioned and stated to support the proposed model in this work; however, this conclusion is not adequately supported by the description and discussions of the model (sections 4.2 to 4.4). 

The authors could consider qualitatively mention the magnetic reconnection as an alternative model, which naturally occurs in turbulent magnetic field, produces plasma cells of all sizes, and accelerate particles. 

====
Major: 

L345: The definition of the duty cycle from Romero et al. 1999 is in contradiction with the other definition based on the fraction of exposure, e.g., if two observations with durations 1 h and 9 h were made, and only in the former micro-variability was detected, the Romano definition would yield a 90% duty cycle while the other definition yield 10%. 
The author should discuss the reasoning of the ROM definition and the quantitative difference between the two definitions. 

L91: and if the conditions were appropriate, we made micro-variability observations of the source.:
The authors should elaborate on the conditions for the decision on whether or not to take micro-variability observations. This is of critical importance, as such decisions directly determine the denominator in the duty cycle calculation. It would only be a fair estimation of the duty cycle if the decision is made randomly; otherwise, the duty cycle results could be biased. 

L105: "A more precise definition of micro-variations would be: brightness changes of 0.001-0.01 magnitudes over timescales of hours or minutes either above or below a linear background sampled over the entire night":
This definition of micro-variations does not give a clear maximum timescale. It is not clear whether authors impose a maximum timescale or the timescale is governed by the duration of observation on a given night. This is important as it determines the maximum variability timescale (and hence the maximum cell size, in the theory proposed by the authors) probed by the micro-variability studies. 
It is unclear that whether or not it is possible that longer variations that fail to meet the definition of micro-variability could be related to the same turbulence (from larger turbulent cells) in the proposed model. If so, the estimation of the duty cycle and other physical parameters would be invalidated. The paper would be strengthened if this caveat, along with the relation between micro-variability and variability of longer timescales, is discussed (i.e., is there a difference in the underlying physical processes, e.g., micro-variability is from turbulent flow and longer variability is from laminar flow, or could they both be from the same turbulent process).  

L119: It would be informative if the sample selection criteria is described. How are the 15 sources selected? Are they based on optical or gamma-ray flux? Or are they the only blazars observed by the SARA observatories?

Figure 5 should include the data points and error bars. 

Table 2: (see earlier general comments) for sources with small numbers of observations, the duty cycle estimation is unreliable. 

Section 4.1 and 3: The authors should consider providing the confidence interval of the bi-serial correlation coefficients, which will statistically strengthen the conclusion of the lack of correlation between brightness and the presence of micro-variability. This could easily be obtained in R. 

L407: "The PDS of the individual light curves did not consistently show a purely noise spectrum":
This is contradictory to the cell size distribution which follows a Kolmogorov distribution. The authors should discuss the relation between the power spectral density and the cell size distribution. 

The formatting needs to be uniform, e.g., L41 has an inline citation [10] without an author name, while L42 has one with the author name. 

Some figures are hard to read (e.g., Figure 2, due to low resolution and small label font size). 

====
Minor: 
L12/86: the full name of SARA is not defined in the text
L25: is an important -> is important
L38: ([3], [4], [5],[6], [7]. : missing space after [5], and missing a closing bracket
L91: VRI should be in italic 
L120: the type of object and the classification is the same column, delete either
L164: from R = 15.4 (0.03) down to R = 19.9 (0.6): what are the numbers in the parentheses? 
L166: with a duty cycle of 22.42: missing %, while text elsewhere used %. Since the duty cycle is already mentioned early in the text, it would be better to move the formal definition (L342 Section 4) to Section 2, or at least reference section 4 when the duty cycle results are first reported. 
L199: FT analysis: FT not defined
L201-202: a maximum rise time of about 0.005 magnitudes per minute: change to something along the line of "a maximum rate of flux rise of about 0.005 magnitudes per minute"
L319: between B ~14 and B ~14?
L405: No consistent underlying periods were f in the micro-variations: what is f? found?
L444: the variables in the equation are not defined. 
L478: a goodness of fit greater than 95%: the goodness of fit is not defined. (is it reduced chi-square?)
L546: why change to the correlation of coefficient instead of "goodness of fit" for BL Lac? 
L566: However, it also failed to show swings in the polarization direction and percentage, consistent with our model: 
There was no mention of polarization in Sections 4.2 to 4.4 where the turbulent jet model was described. It is unclear how and why the lack of polarization swing is consistent with the proposed model. Note that the Marscher 2014 reference (appeared in L44) proposed a turbulent jet model in which polarization swings were expected with flares. 
L589: see above; it is unclear how the proposed model "offers a natural explanation to the erratic swings in polarization direction and degree" 

Author Response

i have uploaded a file with detailed responses.

Reviewer 3 Report

Please excuse my delay in finishing the review. I have attached my report here. 

Author Response

I have uploaded a file with detailed responses.

Round 2

Reviewer 1 Report

I have reviewed the manuscript "The Nature of Micro-variability in Blazars" for the second time. The paper now is much better organized, reads much better, and my comments were adequately addressed. I have only two minor comments before I can fully recommended for publication.

Section 3, 3C454.3, DC is referenced before it is defined.

Section 4.2 PDS--> do the authors mean PSD i.e., power spectral density?

Author Response

 Suggestions for Authors

I have reviewed the manuscript "The Nature of Micro-variability in Blazars" for the second time. The paper now is much better organized, reads much better, and my comments were adequately addressed. I have only two minor comments before I can fully recommended for publication.

Section 3, 3C454.3, DC is referenced before it is defined.  FIXED added in introduction: The duty cycle (DC) is a measure of how close to critical the flow is. The Duty Cycle is defined as the time it shows micro-variations (time on) divided by the total observation time.

Section 4.2 PDS--> do the authors mean PSD i.e., power spectral density?  FIXED

Reviewer 3 Report

My further comments are attached

Author Response

Added section on Fourier analysis at line 44. 

Fourier transform analysis, unequal interval periodic analysis, and wavelets failed to find significant periods, patterns, or noise components (1/f or red noise) [8], [9].  

line 152. Long micro-variability light curves extend up to 10 or 12 hours while shorter ones may only be 3 hours long. Long micro-variability light curves extend up to 10 or 12 hours while shorter ones may only be 3 hours long.

line 497: added text.  Our one dimensional time series cannot specify where in the 3-D jet the individual cells are located, just the arrival times of the pulses.  Future work is envisioned to take 3-D simulated jet turbulence and run a simulated shock wave through it to see the light curves we get from it look like our micro-variability curves (50).  

The final comment on the time series analysis is best seen in the references.  When we did FT and time series analysis we compared our results to simulated data with the same time spacing and randomnly generated amplitudes.  We also used a 1-D clean routine to get our alias, we did our time series analysis very carefully fully aware of problems introduced by uneven sampling.

The lack of these noise components is what led to the model as we specified.

Also the cell sizes deduced and presented in Figure 7 is the strongest evidence that turbulence is the underlying cause of the microvariability, not just the great fits to the individual pulses and the frequency dependence.

We already pointed out the pulse fitting profiles is unique to our model and is very indicative to a turbulent process.  Not sure what lognormal signatures you are referring to as we found no such distribution in the PSD.

I did want to personally thank you and the other two referees because your efforts improved my manuscript 100%.  Thanks very much.  Dr. Webb